# Breastfeeding Practices, Infant Formula Use, Complementary Feeding and Childhood Malnutrition: An Updated Overview of the Eastern Mediterranean Landscape

**DOI:** 10.3390/nu14194201

**Published:** 2022-10-09

**Authors:** Carla Ibrahim, Khlood Bookari, Yonna Sacre, Lara Hanna-Wakim, Maha Hoteit

**Affiliations:** 1Doctoral School of Sciences and Technology (DSST), Lebanese University, Hadath 1533, Lebanon; 2Faculty of Public Health, Section 1, Lebanese University, Beirut 6573, Lebanon; 3PHENOL Research Group (Public Health Nutrition Program Lebanon), Faculty of Public Health, Lebanese University, Beirut 6573, Lebanon; 4Lebanese University Nutrition Surveillance Center (LUNSC), Lebanese Food Drugs and Chemical Administrations, Lebanese University, Beirut 6573, Lebanon; 5Department of Nutrition and Food Sciences, Faculty of Arts and Sciences, Holy Spirit University of Kaslik (USEK), Jounieh P.O. Box 446, Lebanon; 6National Nutrition Committee, Saudi Food and Drug Authority, Riyadh 11451, Saudi Arabia; 7Department of Clinical Nutrition, Faculty of Applied Medical Sciences, Taibah University, Madinah 42353, Saudi Arabia; 8Department of Agricultural and Food Engineering, School of Engineering, Holy Spirit University of Kaslik (USEK), Jounieh P.O. Box 446, Lebanon

**Keywords:** nutritional status, infant and young child feeding, malnutrition, under five years children, Eastern Mediterranean Region

## Abstract

Background: With increasing global rates of overweight, obesity and non-communicable diseases (NCDs) along with undernutrition and micronutrient deficiencies, the Eastern Mediterranean Region (EMR) is no exception. This review focuses on specific nutrition parameters among under five years children, namely ever breastfed, exclusive breastfeeding, mixed milk feeding, continued breastfeeding, bottle feeding, introduction of solid, semi-solid, or soft foods and malnutrition. Methodology: PubMed, Google Scholar, United Nations International Children’s Emergency Fund (UNICEF) databases, World Health Organization (WHO) databases, the World Bank databases and the Global Nutrition Report databases were explored between 10 January and 6 June 2022, to review the nutrition situation among under five years children in the EMR. Results: The regional average prevalence of ever breastfed, exclusive breastfeeding, mixed milk feeding, continued breastfeeding, bottle feeding, introduction of solid, semi-solid, or soft foods was estimated at 84.3%, 30.9%, 42.9%, 41.5%, 32.1% and 69.3%, respectively. Iran, Iraq, Libya and Palestine have seen a decline over time in the prevalence of exclusive breastfeeding. Lebanon, Egypt, Kuwait and Saudi Arabia reported early introduction of infant formula. Moreover, Lebanon, Pakistan, Saudi Arabia and United Arab Emirates were seen to introduce food early to the child, at between 4–6 months of age. The estimated weighted regional averages for stunting, wasting and underweight were 20.3%, 8.9% and 13.1%, respectively. Of concern is the increasing prevalence of stunting in Libya. As for overweight and obesity, the average prevalence was reported to be 8.9% and 3%, respectively. Lebanon, Libya, Kuwait and Palestine showed an increased trend throughout this time. Conclusions: In this review, the suboptimal infant and young child feeding patterns and the twofold incidence of malnutrition in the EMR are highlighted and we urge the prioritizing of measures to improve children’s nutrition.

## 1. Introduction

With eight years remaining to end all forms of malnutrition in accordance with the Sustainable Developmental Goals (SDGs) (Goals 2 and 3) and despite hopes that the world would heal more quickly from the COVID-19 pandemic’s sequelae, still more than half of all infants under six months of age globally are not exclusively breastfed: 22% of children under five years of age are stunted, 6.7% are wasted, 5.7% are overweight and almost 3.1 billion people could not afford a healthy diet, where 78 million in Asia were unable to afford this diet in 2020 [1]. Ensuring that infants and toddlers during the 1000 first days of life receive optimal nutrition is essential for healthy growth, development and metabolic programming during childhood and for promoting health and disease prevention across the life span [2]. A revealing sign of inadequate infant feeding practices worldwide is the enormous global burden of malnutrition [2]. Maternal health and nutrition, infant feeding patterns and nutritional intake, are all variables that modulate malnutrition[3,4]. Malnutrition is defined by insufficient or excessive nutrient intake, an unbalanced intake of vital nutrients, or poor nutrient use. Undernutrition (wasting, stunting, underweight), overweight and obesity are part of the double burden of malnutrition, as are diet-related noncommunicable diseases [5]. Children who experience chronic undernutrition and overweightness early in life fail to reach their maximum growth and development, both physically and mentally and are more susceptible to non-communicable diseases (NCDs) later in life [6]. Moreover, suboptimal infant feeding patterns and infant malnutrition has been associated with increased severity and frequency of infections, raising energy requirements, while reducing appetite and nutrition absorption [1,2,3,4,5,6,7], as well as poor brain development, which can affect school performance and has negative effects on long term careers [8]. However, breastfeeding has a positive correlation with cognitive development and early brain functioning compared to formula-feeding [9].

With increasing global rates of overweight, obesity and NCDs along with undernutrition, micronutrient deficiencies and other forms of malnutrition, the Eastern Mediterranean Region (EMR) is no exception [10]. According to the WHO Eastern Mediterranean (EM) Regional Strategy on Nutrition 2020–2030, the prevalence of undernutrition and NCDs linked to poor dietary intake is still rising [11]. The EM Regional Strategy on Nutrition 2020–2030 set a new plan of action in order to achieve global and regional targets by 2030, including: reduce the number of stunted children under the age of five by half, reduce and maintain childhood wasting to less than 3%, reduce the prevalence of overweight in children under the age of five to less than 3% and improve the rate of exclusive breastfeeding in the first 6 months up to at least 70% [11]. However, despite the clear advantages of breastfeeding, a number of countries in the EMR have reported low rates of exclusive breastfeeding. According to a recent Integrated food security Phase Classification (IPC) analysis of acute malnutrition in Yemen in 2020 [12], the rates of exclusive breastfeeding are far from meeting the World Health Assembly’s 2030 target of having at least 70% of infants under 6 months of age exclusively breastfeeding [11]. Unfortunately, the marketing and promotion of infant formula are key contributing factors to the low rates of exclusive breastfeeding in many countries of the EMR. Besides, bottle feeding immediately after birth was linked to early introduction of solid, semi-solid, or soft foods at 4 months of age [11].

In light of this, our review aimed (1) to analyze the nutrition parameters among under five year old children in the EMR, including ever breastfed, exclusive breastfeeding, mixed milk feeding, continued breastfeeding, bottle feeding, introduction of solid, semi-solid, or soft foods and malnutrition (undernutrition: stunting, wasting, underweight and overnutrition: overweight and obesity) and (2) to compare the findings obtained in this review with other international data on infant feeding patterns and malnutrition among under five children.

## 2. Materials and Methods

### 2.1. Search Strategy

A detailed literature search was conducted between 10 January and 6 June 2022, to review the nutrition situation among under five years children in the EMR, using the following electronic databases: PubMed, Google Scholar, Scopus, United Nations International Children’s Emergency Fund (UNICEF) databases, World Health Organization (WHO) databases, the World Bank databases and the Global Nutrition Report databases. The following search terms were used: “infant feeding pattern”, “breastfeeding”, “complementary feeding”, “malnutrition”, “overweight”, “obesity”, “under-5 children”, “Eastern Mediterranean region”, “Afghanistan”, “Bahrain”, “Djibouti”, “Egypt”, “Iraq”, “Iran”, “Jordan”, “Kuwait”, “Lebanon”, “Libya”, “Morocco”, “Oman”, “Pakistan”, “Palestine”, “Qatar”, “Saudi Arabia”, “Somalia”, “Sudan”, “Syrian Arab Republic”, “Tunisia”, “United Arab Emirates” and “Yemen”. The WHO Regional Office for the Eastern Mediterranean’s most recent classification of EMR countries [13], which now includes the 22 countries mentioned above, guided the selection of these keywords. The keywords were combined in several ways to locate pertinent articles.

### 2.2. Type of Studies and Participants

National published data and English-language publications were included in the search. We also included cross-sectional studies, longitudinal studies, retrospective and prospective, cohort studies and reports. However, reviews and systematic reviews were excluded. Additionally, our target population was constituted of healthy children under the age of five years living in one of the EMR countries listed previously. Studies enrolling children aged above 5 years and having one or more chronic diseases (cardiovascular diseases, diabetes, cancer, chronic kidney diseases and others), were excluded from our search. This review included 326,299 children under the age of 2 for the analysis of the infant and young child feeding practices and 476,928 children under 5 years of age for the analysis of malnutrition parameters; hence, we expect a higher number of children than that reported. These numbers were derived only from the national studies included in this review; however, many other data retrieved from the United Nations International Children’s Emergency Fund (UNICEF) databases, the World Health Organization (WHO) databases, the World Bank databases and the Global Nutrition Report databases did not include the number of children.

### 2.3. Type of Outcomes Reviewed

For infant feeding practices and malnutrition among under five years children, the following parameters proposed by WHO and UNICEF were reviewed. Breastfeeding parameters included: ever breastfed (0–23 months), exclusive breastfeeding under 6 months, mixed milk feeding under 6 months and continued breastfeeding (12–23 months). The prevalence of bottle feeding under 6 months of age was also reported. Complementary feeding parameters included: introduction of solid, semi-solid, or soft foods (6–8 months). Malnutrition parameters among children under the age of 5 years included 2 broad groups: (1) undernutrition: stunting, wasting and underweight, defined as height-for-age z score < −2 standard deviations (SD), weight-for-height z score < −2 SD, weight-for-age z score < −2 SD, respectively, (2) overnutrition: overweight and obesity, defined as weight-for-height z score >2 SD and weight-for-height z score >3 SD, respectively [14].

### 2.4. Data Extraction

To identify papers that were appropriate and pertinent to the review’s objectives, the reviewers first scanned the titles and abstracts of the citations. The data were searched between the years 1988 and 2022. Further, for the articles that seemed to be relevant, the full text study report was obtained and the data was extracted using an Excel template, which included the following details: study number, study title, authors’ name, year of publication, type of study, country, study objectives, study design and participants, study variables and tools, statistical test used and pertinent findings. The selection process is shown in Figure 1. Based on the information available, the weighted regional average for each nutrition parameter was determined. Additionally, the average annual rate of reduction (AARR) for particular nutrition parameters was estimated when data were available at different time points. The methodology for calculating the AARR can be found in the UNICEF technical report [15].

## 3. Results

### 3.1. Infant and Young Child Feeding Practices in the EMR

#### 3.1.1. Breastfeeding Parameters

##### Prevalence of Ever Breastfed

The weighted average prevalence of ever breastfed in the EMR was estimated at 84.3% (Table 1). Ever breastfed prevalence was higher than 95% in United Arab Emirates (95.3%) [16,17], Libya (95.6%) [18], Qatar (96%) [19,20,21], Sudan (96%) [22], Pakistan (96.8%) [23,24,25], Egypt (97.7%) [26,27,28], with the highest rate reached in Iran (98.6%). The lowest rates of ever breastfed were reported from Oman (47%) [29]. Over time, an increasing trend in ever breastfed prevalence was noticed in Lebanon [4,30,31], Egypt [26,27,28] and Kuwait [32,33,34] (Appendix A). As for Afghanistan, Bahrain, Djibouti, Palestine, Somalia and Syrian Arab Republic, no data was reported concerning ever breastfed prevalence.

##### Prevalence of Exclusive Breastfeeding (under 6 Months)

The regional average for exclusive breastfeeding was estimated at 30.9% (Table 1), with the lowest rates being observed in Yemen (9.7%) [35,91], Djibouti (12.4%) [35,39] and Tunisia (13.5%) [35,89]. The highest rates were reported from Afghanistan (57.5%) [35,36], Sudan (55.6%) [35,85,86,87], Iran (46%) [35,46,47,48,49,50], Palestine (43.8%) [35,75,76,77,78], Morocco (42.4%) and Libya (41.3%) [18,67]. As infants grow, the rate of exclusive breastfeeding decreased. In Lebanon, for instance, the prevalence of exclusive breastfeeding decreased from 62.2% among infants aged 0–1 month to 16.54% among those aged 4–6 months at 2016 [61]; similarly, at 2012, the rate of exclusive breastfeeding decreased from 41.5 at 40 days to 12.3% at 6 months [31]. In addition, a Lebanese study conducted in 2017 showed that the proportion of exclusively breastfed infants declined from 40% at age 2 months to 2% at 4–6 months [62]. Many other EMR countries showed similar trend such as Jordan (at birth: 51%, at 1 month: 47%, at 6 months: 33%) [56], Iran (at discharge:74.3%, at 6 months: 28%) [46], Pakistan (at 3 months: 74.4%, at 6 months: 30.2%) [72], Qatar (at 1–3 months: 40%, at 4–6 months: 20%) [19], Saudi Arabia (at birth: 88.6%, at 1 month: 49%, at 2 months: 36.1%, at 4 months: 20.5%, at 6 months: 10.2%) [80] and United Arab Emirates (at 0–3.9 months: 45%, at 0–6 months: 37%, at 4–5.9 month: 26%) [16]. Additionally, over time, some countries in the region have seen a decline in the prevalence of exclusive breastfeeding. This is particularly true for Iran (61% in 2007 and 28% in 2015) [46,48], Iraq (39.7% in 2012 and 25.8% in 2018) [51,52], Libya (44.6% in 1988 and 38% in 2017) [18,67] and Palestine (69.7% in 2007 and 38.9% in 2020) [75,76]. In contrast, an increasing trend was observed in Lebanon (12.3% in 2011 and 59.1% in 2021) [4,31], Pakistan (37.1% in 2006 and 47.8% in 2018) [25,73], Qatar (18.9% in 2009 and 20% in 2017) [19,21], Saudi Arabia (10.2% in 2005 and 28% in 2019) [80,81], Somalia (9% in 2006 and 33.7% in 2018) [83,84] and Sudan (54.6% in 2014 and 62.3% in 2019) [85,87] (Appendix A).

##### Prevalence of Mixed Milk Feeding (under 6 Months)

The average prevalence of mixed milk feeding in the EMR was estimated at 42.9% (Table 1). Mixed milk feeding prevalence was the highest in Iran (87%) [46]. The lowest rates were reported from Qatar (25.6%) [20], Iraq (26.1%) [51] and Libya (26.5%) [18]. An estimate mixed milk feeding prevalence ranged between 29% and 56% was reported among Lebanon (29.3%) [4,30,31,63], Morocco (39.4%) [68], Palestine (41%) [78], Saudi Arabia (42.3%) [80,82], Jordan (43%) [56], United Arab Emirates (48%) [16], Kuwait (50.4%) [32,34,60] and Egypt (55.9%) [27,40,41]. As for the remaining countries: Afghanistan, Bahrain, Djibouti, Oman, Pakistan, Somalia, Sudan, Syrian Arab Republic, Tunisia and Yemen, no data were reported concerning the mixed milk feeding prevalence.

##### Prevalence of Continued Breastfeeding (between 12–23 Months)

The estimated regional average for continued breastfeeding was reported to be 41.5% (Table 1). The lowest rates were seen for Kuwait (12%) [60], Saudi Arabia (11.1%) [80] and United Arab Emirates (1%) [16]. However, the highest rates were reported from Afghanistan (73.8%) [37], Iran (70.5%) [47,50], Sudan (68.2%) [37,85,86,87], Pakistan (65.1%) [23,37,72,73,74] and Oman (64.1%) [29,70,71]. The prevalence of continued breastfeeding decreased the most at 2 years of age. This trend applies to Lebanon (at 1 year: 34.7%, at 2 years: 10.6%) [64], Jordan (at 1 year: 36.2%, at 2 years: 14.9%) [57], Egypt (at 1 year: 80%, at 2 years: 20.4%) [42], Iran (at 1 year: 84.2%, at 2 years: 51%) [47], Iraq (at 1 year: 44.8%, at 2 years: 26.7%) [52], Kuwait (at 1 year: 22%, at 26 months: 2%) [60], Morocco (at 1 year: 64.9%, at 2 years: 29.7%) [69], Oman (at 1 year: 79%, at 2 years: 51%) [70], Pakistan (at 1 year: 68.4%, at 2 years: 56.5%) [74], Palestine (at 1 year: 52.9%, at 2 years: 11.5%) [75], Qatar (at 1 year: 65%, at 2 years: 31.9%) [79], Somalia (at 1 year: 60.8%, at 2 years: 26.8%) [84], Sudan (at 1 year: 89.4%, at 2 years: 48.8%) [87], Syrian Arab Republic (at 1 year: 55.8%, at 2 years: 24.9%) [88], Tunisia (at 1 year: 45.4%, at 2 years: 18.2%) [89] and Yemen (at 1 year: 71.2%, at 2 years: 45.3%) [91] (Appendix A). No data were reported on the prevalence of continued breastfeeding in Bahrain, Djibouti and Libya.

##### Prevalence of Bottle Feeding (under 6 Months)

The average prevalence of bottle feeding in the EMR was estimated at 32.1% (Table 1). Saudi Arabia reported the highest rate (59.2%)[80], followed by Oman with a bottle feeding prevalence of 48.6% [29,70]. The lowest rates were shown in Kuwait (16.7%) [18]. Early introduction of infant formula was reported in some countries of the region, such as in Lebanon [4,31,61,63], Egypt [26,40,41], Kuwait [32,34,60] and Saudi Arabia [80]. There is a scarcity of data in the literature on bottle feeding prevalence in many countries of the Region. This is accurate for Afghanistan, Bahrain, Djibouti, Jordan, Qatar, Somalia, Sudan, Syrian Arab Republic, United Arab Emirates and Yemen.

#### 3.1.2. Complementary Feeding Parameters

##### Prevalence of Introduction of Solid, Semi-Solid or Soft Food (at 6–8 Months)

The regional average prevalence for introduction of solid, semi-solid or soft food was estimated at 69.3% (Table 1), with the lowest rates being observed in Saudi Arabia (14.3%) [80], Somalia (41.2%) [37,84] and Lebanon (43.6%) [4,30,31,63,65]. The highest rates were reported from Oman (95.3%) [29,37,71], United Arab Emirates (93.5%) [16,17], Palestine (89.9%) [37,75], Tunisia (86.8%) [37,89], Morocco (84.4%) [37,69], Jordan (83.4%) [37,57] and Iraq (82.5%) [52,53]. Some EMR countries were seen to introduce food early to the child, between 4–6 months of age. This is particularly accurate for Lebanon (37%) [4], Pakistan (55%) [24], Saudi Arabia (81.5%)[80] and United Arab Emirates (19%) [16]. Besides, another trend of introduction of solid, semi-solid or soft food earlier than 4 months of age was observed at lower rates among Lebanon (1.3%) [63], Saudi Arabia (4.2%) [80] and United Arab Emirates (7%) [16] (Appendix A). However, no data were reported on the prevalence of introduction of solid, semi-solid or soft food in Bahrain, Djibouti, Kuwait and Libya.

### 3.2. Malnutrition Status among Under-5 Years Children in the EMR

#### 3.2.1. Malnutrition Parameters

##### Undernutrition: Prevalence of Stunting, Underweight and Wasting

The estimated weighted regional averages for stunting, wasting and underweight were 20.3%, 8.9% and 13.1%, respectively (Table 2). The prevalence of wasting was highest in Djibouti (21.5%) [39,92], Yemen (16.3%) [91,92,93], Somalia (15.4%) [83,84,92,94], Sudan (15.1%) [48,49,50,92,95], Pakistan (13.8%) [73,74,92,96,97] and Afghanistan (12.8%) [36,92,98], while the prevalence of underweight was highest in Yemen (39%) [93], Pakistan (35.6%) [74,96,97,99] and Somalia (29.3%) [83,99]. The incidence of wasting has increased over time in countries with political instability, such as Egypt (3.3% in 1992 vs. 9.5% in 2014) [92,100] and Libya (3.7% in 1995 vs. 10.2% in 2014) [92,101]. Similarly, an increasing trend in the prevalence of underweight was observed in Iran (in 2009: 4.8% vs. in 2018: 7.63%) [102,103] and Pakistan (in 2014: 33.2% vs. in 2017/18: 57.3%) [96,97] (Appendix A). The highest rates of stunting were noted in Pakistan (46.8%) [73,74,96,97,104], Yemen (43.5%) [91,93,104] and Somalia (29.3%) [83,84,94,104]. Of concern is the increasing prevalence of stunting that has been observed in Libya (in 2014: 38.1% vs. in 2020: 43.5%) [104,105] (Appendix A). The AARR for the prevalence of stunting in the region was estimated at 2.5%.

##### Overnutrition: Prevalence of Overweight and Obesity

The estimated weighted average prevalence of overweight and obesity in children aged under five years was 8.9% and 3%, respectively (Table 2). The highest prevalence of overweight was observed in Libya (23.7%) [101,105,106], the Syrian Arab Republic (18.1%) [88,106] and Tunisia (16.9%) [89,106], while the lowest was reported from Yemen (2.6%) [91,106], Somalia (3%) [84,106], Sudan (3.4%) [85,87,106] and Afghanistan (4%) [36,106]. As for obesity, the highest rate was reported in Bahrain (6.5%) [107], yet Sudan has the lowest rate (0.9%) [85], followed by Iran (1.3%) [108] and Palestine (1.5%) [109]. According to the data currently available, the prevalence of overweight and obesity throughout this time appears to be on the rise. This was particularly true for Lebanon (overweight: in 2011: 6.5% vs. in 2021: 16.8%; obesity: in 2011: 2.7% vs. in 2021: 8.9%) [4,110], Libya (overweight: in 1995: 16.2% vs. in 2020: 25.4 [105,106], Kuwait (overweight: in 2017: 5.5% vs. in 2020: 7.1%) [106,111] and Palestine (overweight: in 2014: 7.3% vs. in 2020: 8.6%) [75,109] (Appendix A).

### 3.3. International Overview

The global average prevalence of exclusive breastfeeding was shown to be 44% [35]. Besides, the estimated average prevalence of exclusive breastfeeding reported from the North America Region [35], the East Asia and Pacific Region [35] the Latin America and Caribbean Region [35], the South Asia Region [35] and the Europe and Central Asia Region [35] was 26%, 31%, 37%, 57% and 41%, respectively. Regarding continued breastfeeding, the estimated global average prevalence was 65% [37]. As for the East Asia and Pacific Region excluding China [37], the Latin America and Caribbean Region[37], the South Asia Region [37], the Europe and Central Asia Region [37] and the North America Region [37], the prevalence of continued breastfeeding was 58%, 45%, 78%, 50% and 12%, respectively. Globally, the estimate average prevalence of introduction of solid, semi-solid, or soft foods (69.3%) was 73% [37]. This prevalence was reported, among different regions, as follows: the East Asia and Pacific Region (84%) [37], the Latin America and Caribbean Region (87%) [37], the Europe and Central Asia Region (76%) [37], the South Asia Region (58%) [37].

In order to track undernutrition among different regions around the globe, the prevalence of wasting, stunting and underweight was reported. The worldwide average of wasting was 6.7% [92]. In addition, wasting estimates from different international regions were stated as follow: East Asia and the Pacific (3.7%) [92], Latin America and the Caribbean (1.3%) [92], North America (0.2%) [92] and South Asia (14.7%) [92]. As for underweight, the global average prevalence was 12.6% [99]. Furthermore, East Asia and Pacific [99], Latin America and Caribbean [99], North America [99] and South Asia (27.4%) [99], showed a prevalence of 5.2%, 2.7%, 0.7% and 27.4%, respectively. Stunting worldwide average prevalence was estimated to be 22% [104]. The estimated stunting prevalence reported from East Asia and Pacific [104], Latin America and Caribbean [104], North America [104], Central Europe and the Baltics [104], Europe and Central Asia [104] and South Asia [104], was 13.4%, 11.3%, 3.2%, 4.5%, 5.7% and 31.8%, respectively.

The estimated global average for overweight among under five years children was 5.7% [106]. This prevalence was reported, among different regions, as follows: East Asia and Pacific (7.8%) [106], Latin America and Caribbean (7.5%) [106], North America (9.1%) [106], Central Europe and the Baltics (6.6%) [106], Europe and Central Asia (7.9%) [106] and South Asia (2.2%) [106]. As for obesity, the average prevalence reported in Latin America and Caribbean [125] and Central and Eastern Europe and Central Asia [125] was 9.2% and 10.9%, respectively.

## 4. Discussion

This review highlighted the feeding patterns of infant and young children in the EMR. The regional average prevalence of ever breastfed, exclusive breastfeeding, mixed milk feeding, continued breastfeeding, bottle feeding, and introduction of solid, semi-solid, or soft foods was estimated at 84.3%, 30.9%, 42.9%, 41.5%, 32.1% and 69.3%, respectively. Over time, some countries in the EMR region have seen a decline in the prevalence of exclusive breastfeeding, such as Iran, Iraq, Libya and Palestine. Further-more, Lebanon, Egypt, Kuwait and Saudi Arabia reported early introduction of infant formula. Lebanon, Pakistan, Saudi Arabia and United Arab Emirates were also seen to introduce food early to the child, between 4–6 months of age. Additionally, this paper underlined the double burden of malnutrition among under five years children in the EMR, with undernutrition coexisting with overnutrition in most countries. The estimated weighted regional averages for stunting, wasting and underweight were 20.3%, 8.9% and 13.1%, respectively. Of concern is the increasing prevalence of stunting that has been observed in Libya. As for overweight and obesity, the average prevalence was reported to be 8.9% and 0.9%, respectively. Lebanon, Libya, Kuwait and Palestine described an increased trend throughout this time.

### 4.1. Infant and Young Child Feeding Practices

Despite the fact that the majority of children in the EMR were breastfed at some point during their childhood, the regional average prevalence of exclusive breastfeeding during the first six months of life was only 30.9%. This rate was much lower than the global average prevalence of exclusive breastfeeding (44%) [35], lower than the estimated average prevalence reported from the Latin America and Caribbean Region (37%) [35], the South Asia Region (57%) [35] and the Europe and Central Asia Region (41%) [35], higher than that reported from the North America Region (26%) [35], similar to that of the East Asia and Pacific Region (31%) [35] and far from the WHO global goal of 50% exclusive breastfeeding by 2025 and 70% by 2030 [126]. The reported de-cline in exclusive breastfeeding in certain countries of the region is of greater concern. Furthermore, decreased rates of continued breastfeeding, particularly throughout the child’s second year, and increased proportion of bottle feeding and mixed milk feeding were described in the EMR. The prevalence of bottle feeding is highest in Western Europe, Australia and North America regions, but these countries’ rates are stable, while the Middle East and North Africa regions are expected to have the biggest increases [127,128]. The estimated regional average of continued breastfeeding (41.5%) was considerably lower than the global average (65%) [37], the East Asia and Pacific Region excluding China (58%) [37], the Latin America and Caribbean Region (45%) [37], the South Asia Region (78%) [37] and the Europe and Central Asia Region (50%) [37]. However, our findings were higher than the North America Region (12%) [37]. This inadequate adherence to the WHO infant feeding guidelines could have an adverse impact on the disease burden in the EMR. Referring to the literature, it has been demonstrated that healthy and optimal nutrition at earlier stages in the infants’ life may have a critical role in promoting cognitive and physical growth, boosting immunity, reducing the risk of childhood obesity and preventing NCDs [4,129,130]. Poor Baby-friendly Hospital Initiative (BFHI) and Code implementation, the limited knowledge of healthcare professionals in assisting breastfeeding mothers, high rates of pre-lacteal feeding, the absence of designated maternity facilities such as lactation rooms in workplaces and inadequate support for breastfeeding mothers may be accountable for the inadequate status of the aforementioned breastfeeding parameters [131,132,133,134,135]. When it comes to complementary feeding, the majority of EMR infants were introduced to solid, semi-solid, or soft meals at 6–8 months; this was in accordance with the WHO recommendations [126]. Yet our estimate average prevalence of introduction of solid, semi-solid, or soft foods (69.3%) was lower than the global estimated average (73%) [37], the East Asia and Pacific Region (84%) [37], the Latin America and Caribbean Region (87%) [37] and the Europe and Central Asia Region (76%) [37] and higher than the South Asia Region (58%) [37]. However, some EMR countries (Lebanon, Pakistan, Saudi Arabia and United Arab Emirates) were seen to introduce food early to the child. Early introduction of solid, semi-solid, or soft foods has been shown to lower consumption of protective components contained in breastmilk, which may increase newborn morbidity. After introducing solid food, women might consequently produce less breastmilk, which could negatively impact the infant’s intake of nutrients. Additionally, improper handling and storage of complementary foods might expose infants to dangerous microbes [4,136]. Besides, low socioeconomic status, food insecurity and traditions are all significant factors that have an impact on complementary feeding practices [137,138,139,140]. Although some countries of the region showed some progress, all in all a decreasing trend was recorded in the infant and young child feeding parameters; hence, further interventions are needed.

### 4.2. Malnutrition Status among Under-5 Years Children

The results of this research demonstrated that undernutrition among young children continues to be a major problem in a number of countries in the region. According to the WHO cutoffs, only a few countries (Jordan, Kuwait, Morocco, Qatar and Tunisia) had levels of wasting below the recommended threshold of 3% [11], while the regional average (8.9%) was higher. The regional average was also observed to be higher than the worldwide average of wasting (6.7%) [92], as well as estimates from East Asia and the Pacific (3.7%) [92], Latin America and the Caribbean (1.3%) [92] and North America (0.2%) [92], although it was lower than those reported from South Asia (14.7%) [92]. As for underweight, the regional average (13.1%) was lower than that reported in 2018 (18%) [3], indicating that there has been some progress on this front. This regional average was close to the global average (12.6%) [99], higher than that of East Asia and Pacific (5.2%) [99], Latin America and Caribbean (2.7%) [99] and North America (0.7%) [99], while it was lower than that reported from South Asia (27.4%) [99]. Additionally, a rise in the prevalence of stunting was noted in Libya, underlining the country’s growing vulnerability to chronic undernutrition. Stunting prevalence in the region was assessed to have an AARR of 2.5%, which was lower than the rate required to meet the global nutrition target set by the World Health Assembly (AARR = 3.9%) [141]. Actually, Lebanon (AARR = 4.5%), Afghanistan (AARR = 4.1%), Iraq (AARR = 10.4%), Morocco (AARR = 4.5%), Palestine (AARR = 7.2%) and Saudi Arabia (AARR = 5.9%) seemed to be making some progress toward achieving the target. The regional average of stunting (20.3%) exceeded estimates reported from East Asia and Pacific (13.4%) [104], Latin America and Caribbean (11.3%) [104], North America (3.2%) [104], Central Europe and the Baltics (4.5%) [104], and Europe and Central Asia (5.7%) [104], although it was lower than that of South Asia (31.8%) [104] and close to the worldwide average of 22% [104]. Stunting, wasting and underweight rates have been declining in some countries of the region and this trend may be attributed to a number of factors, including higher levels of maternal education, a gradual rise in the number of women’s and children’s health centers, a higher percentage of women receiving antenatal care, and increased vaccination rates [142,143]. However, conflicts, economic and political instability and the COVID-19 pandemic together led to no progress and even an increase in this trend, in other countries of the EMR [3,4].

Overweight and obesity in under five years children in the region are of utmost concern. Among pre-school aged children, the prevalence of overweight/obesity increased in several countries, including Lebanon, Libya, Kuwait and Palestine. The estimated regional average for overweight among under five years children was 8.9%, which exceeds the worldwide average (5.7%) [106] and that reported for East Asia and Pacific (7.8%) [106], Latin America and Caribbean (7.5%) [106], North America (9.1%) [106], Central Europe and the Baltics (6.6%) [106] and Europe and Central Asia (7.9%) [106], but it was lower than estimates reported for South Asia (2.2%) [106]. As for obesity, the average regional estimate (3%) was lower than the average prevalence reported in Latin America and Caribbean (9.2%) [125] and Central and Eastern Europe and Central Asia (10.9%) [125]. Juvenile obesity can cause adverse health effects later on in children’s life [144,145,146]. Higher and lower socioeconomic status, parental obesity, sedentary lifestyle, high intake of sugar and fat rich food and low adherence to a balanced and healthy diet have been identified as risk factors for childhood obesity [147,148,149,150]. In order to promote better diets as part of effective obesity prevention and nutrition policies, the focus should be on unhealthy diets and feeding patterns and, more specifically, on the requirement to restrict the marketing of food and beverages to children [151].

### 4.3. Strength and Limitations

Our review provides updated data on the nutritional situation among under five years children in the EMR; it sheds important light on infant and young child feeding parameters including ever breastfed, exclusive breastfeeding, mixed milk feeding, continued breastfeeding, bottle feeding, introduction of solid, semi-solid, or soft foods, as well as malnutrition parameters including undernutrition: wasting, stunting, under-weight and overnutrition: overweight, obesity. However, this study has some limitations. In many cases, the lack of recent, nationally representative research assessing the nutritional status of the pre-school children population in several countries in the region and the lack of studies examining long-term trends in nutritional parameters limited the data on nutritional parameters that were available.

## 5. Conclusions

Inadequate infant feeding practices is one of the leading factors contributing to the increased rates of malnutrition in all of its various form. In light of this, the thorough search conducted in this study urges prompt action to address the suboptimal feeding patterns and the double burden of malnutrition among under five years children in the EMR. Therefore, the improvement of the population’s nutritional status should be a top priority, especially for infants and young children.

## Figures and Tables

**Figure 1 nutrients-14-04201-f001:**
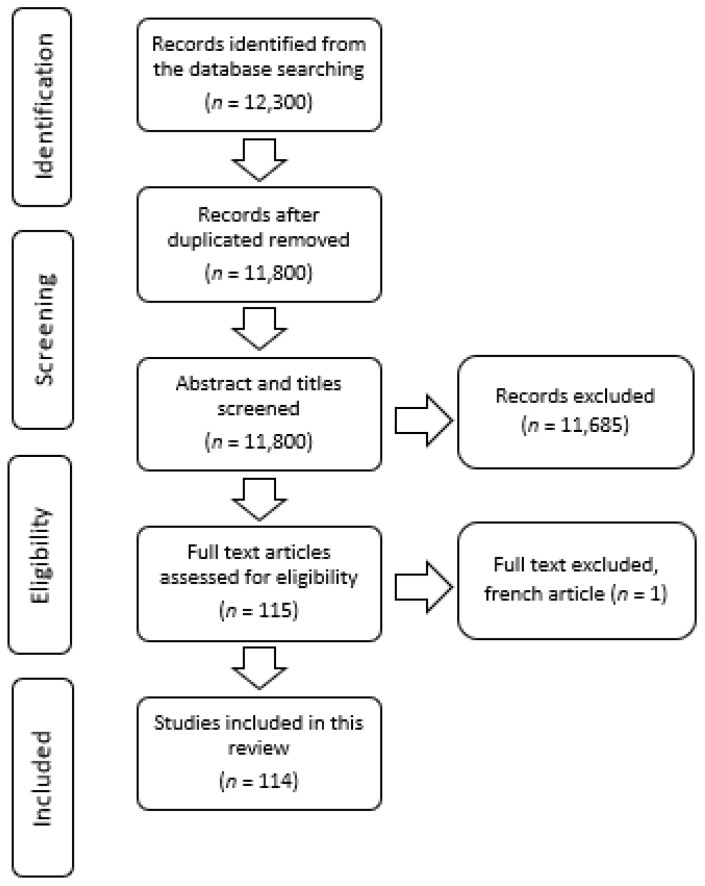
Search flow chart illustrating the number of studies and reports evaluated and subsequently included in the review. (*n* = number of studies).

**Table 1 nutrients-14-04201-t001:** Prevalence of infants and young children feeding patterns in the EMR.

Eastern Mediterranean Countries	Number of Children *	Ever Breastfed (%)	Exclusive Breastfeeding (%)	Mixed Milk Feeding (%)	Continued Breastfeeding (%)	Bottle Feeding (%)	Introduction of Solid, Semi-Solid, or Soft Foods (%)	References
Afghanistan (2015–2018)	7963	NA	57.5	NA	73.8	NA	58.5	[35,36,37,38]
Bahrain	NA	NA	NA	NA	NA	NA	NA	NA
Djibouti (2012–2017)	ND	NA	12.4	NA	NA	NA	NA	[35,39]
Egypt (2010–2018)	7331	97.7	29.1	55.9	50.3	34	72.3	[26,27,28,35,37,40,41,42,43,44,45]
Iran (2005–2015)	93,627	98.6	46	87	70.5	31.2	75.9	[35,37,46,47,48,49,50]
Iraq (2006–2018)	1598	88.5	30.1	26.1	35.4	32.6	82.5	[35,37,51,52,53,54,55]
Jordan (2014–2017)	1444	84.4	21.1	43	25.8	NA	83.4	[35,37,56,57,58,59]
Kuwait (2007–2015)	2726	93.6	26.5	50.4	12	16.7	NA	[32,33,34,60]
Lebanon (2000–2021)	9481	89.7	25.4	29.25	24.9	30.3	43.6	[4,30,31,37,61,62,63,64,65,66]
Libya (1988–2017)	526	95.6	41.3	26.5	NA	27	NA	[18,67]
Morocco (2003–2017)	271	94.7	42.4	39.4	41.2	30.3	84.4	[35,37,68,69]
Oman (2016–2017)	1344	47	24.6	NA	64.1	48.6	95.3	[29,37,70,71]
Pakistan (2006–2018)	26,872	96.8	37.2	NA	65.1	19.5	52.5	[23,24,25,37,72,73,74]
Palestine (2003–2020)	835	NA	43.8	41	30.4	23.2	89.9	[35,37,75,76,77,78]
Qatar (2009–2017)	1418	96	24.4	25.6	36.5	NA	74	[19,20,21,35,37,79]
Saudi Arabia (2004–2019)	16,258	91.6	21.9	42.3	11.1	59.2	14.3	[80,81,82]
Somalia (2006–2018)	ND	NA	25.5	NA	44.2	NA	41.2	[35,37,83,84]
Sudan (2014–2019)	152,259	96	55.6	NA	68.2	NA	54.7	[22,35,37,85,86,87]
Syrian Arab Republic (2006–2019)	ND	NA	28.5	NA	42.5	NA	74.6	[35,37,88]
Tunisia (2018)	ND	NA	13.5	NA	NA	16.7	86.8	[35,37,89]
United Arab Emirates (2014–2020)	2346	95.3	32.7	48	1	NA	93.5	[16,17,90]
Yemen (2013)	ND	NA	9.7	NA	60.5	NA	69.2	[35,37,91]
Eastern Mediterranean Region	326,299							
Point Estimate		84.3	30.9	42.9	41.5	32.1	69.3	
Lower Limit		47	9.7	25.6	1	16.7	14.3	
Upper Limit		98.6	57.5	87	73.8	59.2	95.3	

NA: not available; ND: not determined. * The number of children was derived from the national published studies.

**Table 2 nutrients-14-04201-t002:** Prevalence of malnutrition in the EMR.

Eastern Mediterranean Countries	Number of Children *	Stunting (%)	Underweight (%)	Wasting (%)	Overweight (%)	Obesity (%)	References
Afghanistan (2018–2020)	600	36.7	19.1	12.8	4	NA	[36,92,98,99,104,106]
Bahrain (1995–2020)	698	5.1	7.6	6.6	8.2	6.5	[92,99,104,106,107]
Djibouti (2012–2020)	ND	33.8	26	21.5	7.7	NA	[39,92,99,104,106]
Egypt (2008–2020)	5144	22.6	10.4	10.9	12.6	NA	[42,45,92,99,100,104,106,112]
Iran (2004–2020)	84,667	8.7	6.2	5.2	5.2	1.3	[47,92,99,102,103,104,106,108,113,114,115]
Iraq (2006–2020)	2290	25.4	9.9	4.2	7.5	NA	[51,52,92,99,104,106,116,117]
Jordan (2012–2020)	ND	7.5	2.7	2.4	5.9	NA	[57,92,99,104,106]
Kuwait (2016–2020)	4400	6.2	3	2.5	7.8	3.7	[92,99,104,106,111,118]
Lebanon (2004–2021)	11,505	9.7	4.4	6	14.6	4.3	[4,30,63,64,66,92,99,104,106,110]
Libya (1995–2020)	9846	30.8	8	8	23.7	NA	[92,99,101,104,105,106,119]
Morocco (2016–2020)	297	14.4	4.4	2.8	11.1	NA	[68,69,92,99,104,106]
Oman (2016–2020)	3129	11.6	11.2	9.3	4.5	NA	[29,71,92,99,104,106]
Pakistan (2014–2020)	93,159	46.8	35.6	13.8	5.1	NA	[73,74,92,96,97,99,104,106]
Palestine (2003–2020)	10,677	16.2	10.8	9.4	8.1	1.5	[75,77,78,92,99,104,106,109,120,121]
Qatar (1995–2020)	ND	4.6	4.8	2.1	13.9	NA	[92,99,104,106]
Saudi Arabia (2004–2020)	15,516	8	6.1	11.1	6.9	NA	[92,99,104,106,122,123]
Somalia (2006–2020)	73,778	31.4	29.3	15.4	3	NA	[83,84,92,94,95,96,97,98,99,104,106]
Sudan (2010–2020)	146,797	34.2	25.5	15.4	3.4	0.9	[85,86,87,92,95,99,104,106]
Syrian Arab Republic (2010–2020)	ND	28.8	10.4	11.5	18.1	NA	[88,92,99,104,106]
Tunisia (2018–2020)	ND	8.5	1.6	2.1	16.9	NA	[89,92,99,104,106]
United Arab Emirates (2019–2020)	801	12.5	NA	7	5	3	[16,124]
Yemen (2013–2020)	13,624	43.5	39	16.3	2.6	NA	[91,92,93,99,104,106]
tern Mediterranean Region	476,928						
Point Estimate		20.3	13.1	8.9	8.9	3	
Lower Limit		4.6	1.6	2.1	2.6	0.9	
Upper Limit		46.8	39	21.5	23.7	6.5	

NA: not available; ND: not determined. * The number of children was derived from the national published studies.

## Data Availability

The data presented in this study are available in the Appendix A.

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
