# Peer review of "Breastfeeding Practices, Infant Formula Use, Complementary Feeding and Childhood Malnutrition: An Updated Overview of the Eastern Mediterranean Landscape"

_nutrients, 2022, doi:10.3390/nu14194201_

Round 1

Reviewer 1 Report

Dear authors I have to congratulate you for your efforts to deal with a very sensitive and important issue. Before seen your paper published I have raised several concerns and suggestions aiming in improving your manuscript

Line 19. ‘Nutrition indicators’ it is a no correct term here. Please substitute with ie early nutrition documentation or parameters or something similar

Line 20 onwards. Why authors use the age limit of 5 years? The parameters of nutrition that authors attempt to study take place almost exclusively at the first or in few cases the second year of life. Please give an explanation to justify this selection

Line 25. the abbreviations the authors use are somehow odd. I would prefer not to abbreviate these terms but use the full words. This would be extremely helpful to the readers especially those that are not pediatricians.

INTRODUCTION

General comment:  the introduction is long and not well focused. Please write it again having half the original size. References in political situation, covid etc are superfluous and I suggest omitting and staying in an absolutely scientific report

Lines 56-58:  SAM and MAM are strange abbreviations also. Please do not use

You are speaking for pediatric malnutrition in general and later consequences. Please focus on the effects of breastfeeding or formula feeding in later health because this is more compatible with the title and the aim of the present study.

Line  71: obesity and overweight are over nutrition. Why authors introduce another parameter irrelevant with the title of this study? On my opinion it is sufficient the authors search exactly what the title of this study imply and not to attempt to complicate their findings.

MATERIAL AND METHODS

How many years were searched back ? Feeding conditions change over time. This should be posed clearly in this section

line 120  again the selection of children<5 years seems arbitrary . Why not below 10 ? The most reasonable is to study the 2 first years of life as we stated above or all the period of childhood .

line 118:  how the authors decided that one study is ‘National representative?” or it is not despite its qualitative characteristics?

RESULTS

Line 182 onwards

This paragraph should be on a table please convert in tabular data . Please make the same for the other section of the results. It is the nature of this review that needsa tabular data  imperatively.

It would be helpful to know how many studies was found of each area and how many children were studied in each study

The tables in supplementary material need to be improved and become more professional

DISCUSSION

It is in line with the results, please avoid to repeat several things

RECOMENTATIONS

No need for this section especially in a so long and generalized way. One or two lines at the conclusion section would be sufficient

Reviewer 2 Report

Thank you for the opportunity to review this manuscript. The general impression is the following:

1. the paper is extremely long, unclear and difficult to read

2. there are methodological problems: time frame to which the various selected articles refer; definitions of undernutrition, obesity etc. not clearly defined; it is not declared how the quality of the article has been evaluated;

3. The aim of the study: point 2 is not clear (international data) and the data relating to this aim are not shown in the manuscript.

4. The same results are presented several times: in the text, tables and figures.

5. Recommendations must be evidence-based and shared by experienced teams.

Round 2

Reviewer 1 Report

I found this version improved. Thank you

Reviewer 2 Report

The results relating to aim 2 must be reported in the appropriate section, that is, the results section. If the authors wish to discuss the results obtained from the review with other international data then this becomes a discussion issue and no longer a result.

Author Response

The results relating to aim 2 must be reported in the appropriate section, that is, the results section. If the authors wish to discuss the results obtained from the review with other data then this becomes a discussion issue and no longer a result.
Reply: Thank you for addressing this comment. We added accordingly a section in the results part entitled “International overview” where we stated all available malnutrition and infant and young child feeding prevalence among different region around the world. Please refer to line 286 in the manuscript (paragraph is highlighted in yellow).